# In Silico Analysis of miRNA-Mediated Genes in the Regulation of Dog Testes Development from Immature to Adult Form

**DOI:** 10.3390/ani13091520

**Published:** 2023-04-30

**Authors:** Vanmathy R. Kasimanickam, Ramanathan K. Kasimanickam

**Affiliations:** 1Center for Reproductive Biology, College of Veterinary Medicine, Washington State University, Pullman, WA 99164, USA; vkasiman@wsu.edu; 2Department of Veterinary Clinical Sciences, College of Veterinary Medicine, Washington State University, Pullman, WA 99164, USA

**Keywords:** canine, testis, micro RNA, messenger RNA, bioinformatics, biological function

## Abstract

**Simple Summary:**

The objective of this investigation was to elucidate the association of miRNA-mediated genes in the regulation of dog testes development from immature to adult form by in-silico analysis. In silico analysis of differentially expressed (DE) testis miRNAs between healthy immature and mature dogs were performed using miRNet, STRING, and ClueGo programs. The determination of mRNA and protein expressions of predicted pivotal genes and their association with miRNA were studied. The predicted genes are involved in the governing of several key biological functions (cell cycle, cell proliferation, growth, maturation, survival, and apoptosis) in the testis as they evolve from immature to adult forms, mediated by several key signaling pathways (ErbB, p53, PI3K-Akt, VEGF, and JAK-STAT), cytokines and hormones (estrogen, GnRH, relaxin, thyroid hormone, and prolactin). Elucidation of DE-miRNA predicted genes’ specific roles, signal transduction pathways, and mechanisms, by mimics and inhibitors, which could perhaps offer diagnostic and therapeutic targets for infertility, cancer, and birth control.

**Abstract:**

High-throughput in-silico techniques help us understand the role of individual proteins, protein–protein interaction, and their biological functions by corroborating experimental data as epitomized biological networks. The objective of this investigation was to elucidate the association of miRNA-mediated genes in the regulation of dog testes development from immature to adult form by in-silico analysis. Differentially expressed (DE) canine testis miRNAs between healthy immature (2.2 ± 0.13 months; *n* = 4) and mature (11 ± 1.0 months; *n* = 4) dogs were utilized in this investigation. In silico analysis was performed using miRNet, STRING, and ClueGo programs. The determination of mRNA and protein expressions of predicted pivotal genes and their association with miRNA were studied. The results showed protein–protein interaction for the upregulated miRNAs, which revealed 978 enriched biological processes GO terms and 127 KEGG enrichment pathways, and for the down-regulated miRNAs revealed 405 significantly enriched biological processes GO terms and 72 significant KEGG enrichment pathways (False Recovery Rate, *p* < 0.05). The in-silico analysis of DE-miRNA’s associated genes revealed their involvement in the governing of several key biological functions (cell cycle, cell proliferation, growth, maturation, survival, and apoptosis) in the testis as they evolve from immature to adult forms, mediated by several key signaling pathways (ErbB, p53, PI3K-Akt, VEGF and JAK-STAT), cytokines and hormones (estrogen, GnRH, relaxin, thyroid hormone, and prolactin). Elucidation of DE-miRNA predicted genes’ specific roles, signal transduction pathways, and mechanisms, by mimics and inhibitors, which could perhaps offer diagnostic and therapeutic targets for infertility, cancer, and birth control.

## 1. Introduction

MicroRNAs (miRNAs) play a key role in the differentiation, development, maintenance, and functions of various tissues. Spermatogenesis is a sequence of complex processes [1,2], which are regulated by pathways mediated by miRNAs [3,4]. The genome of testicular cells is actively transcribed into RNAs that involves many non-coding RNAs consisting of circular RNAs (circRNAs) and miRNAs to regulate and generate phase-specific gene expression patterns [5,6]. In mouse testis, pachytene, round, and elongated spermatocytes showed the highest levels of miRNA expressions [7,8,9]. Numerous miRNAs are preferentially expressed in the testis and male germ cells of humans and mice [5,6,7,8,9]. However, the biological functions of many miRNAs involved in spermatogenesis and testicular function are largely unknown. It should be noted that diminution of the Dicer gene upsets the proliferation and differentiation of mouse spermatogenic germ cells [10,11,12].

Dicer is an endonuclease enzyme that belongs to the ribonuclease III (RNase III) family. It activates the RNA-induced silencing complex (RISC), which is essential for RNA interference. The RISC contains dsRNA binding proteins, including protein kinase RNA activator (PACT) and transactivation response RNA binding protein (TRBP) that process pre-microRNAs into mature microRNAs (miRNAs) that target specific mRNA species for regulation. Dicer plays an important role in spermatogenesis [13,14]. MiRNAs along with dicer act as a post-transcriptional regulatory unit in testicular tissue development and spermatogenesis [15,16]. Therefore, miRNAs can be targeted for evaluating male fertility and can serve as useful biomarkers.

It is conceivable that miRNAs can regulate meiosis and thus spermatogenesis [17] by regulating mRNA degradation and disrupting mRNA translation [18,19]. While recent studies focused on miRNA tissue expression in rodents and humans, miRNA data for dogs are lacking [20,21]. We recently reported miRNA expression patterns between sexually immature and mature canine testes [20].

The objective of this investigation was to elucidate miRNA–mRNA interaction of differentially expressed (DE) miRNAs between immature and mature canine testis by constructing a protein–protein network and performing cluster gene analysis to elucidate key biological functions. 

## 2. Materials and Methods

### 2.1. miRNA Data Profiling

The DE-miR data used in this investigation were obtained from the testes of healthy Labrador-mix dogs (young: 2.2 ± 0.13 months; *n* = 4 and adult: 11 ± 1.0 months; *n* = 4) undergoing elective castrations [20]. Briefly, RNA was isolated from the testis using the TRIzol homogenization method, and RNA concentration and quality were determined [20,21]. Then, RNA samples were reverse-transcribed using a Micro Script II RT kit (Qiagen, Frederick, MD, USA). Micro Script HiSpec buffer (5X) was used to prepare cDNA for mature miRNA profiling, which was performed using miRNome miScript miRNA polymerase chain reaction (PCR) array kits. The Canine miScript miRNA PCR Array plate (Qiagen) included primers for 84 mature miRNAs and controls (Table 1). The controls were cel-miR-39-3p (H01 and H02), SNORD61 (H03), SNORD68 (H04), SNORD72 (H05), SNORD95 (H06), SNORD96A (H07), RNU6-2 (H08), miRTC (H09 and H10), and PPC (H11 and H12). SNORDs and RNU6-2 served as internal normalizers. Two reverse transcription controls and two positive controls ensured the efficiency of the array, reagents, and instrument performance.

In order to obtain miRNA-predicted gene datasets from young and adult dogs, the miRNet web tool (http://www.mirnet.ca/) (accessed on 4 March 2022) [22,23] was used to analyze retrieved miRNA datasets, and then Protein–Protein Interaction (PPI) networks of co-expressed genes were designed using the Search Tool for the Retrieval of Interacting Genes/Proteins (STRING) online database (http://stringdb.org/) (accessed on 4 March 2022) [24] and ClueGo (accessed on 4 March 2022) [25].

### 2.2. Conserved Nucleotide Sequences

Nucleotide sequences of differentially expressed canine miRNAs were retrieved from miRBase, (www.mirbase.org) (accessed on 4 March 2022) and compared with those of homo sapiens for similarities [22,23].

### 2.3. Identification of Target Genes of Differentially Expressed miRNAs

The target and predicted genes of DE-miRNAs were retrieved using miRNet (http://www.mirnet.ca/) (accessed on 4 March 2022) [24,25]. This tool integrated data from different miR databases (TarBase, miRTarBase, and miRecords) and identified target and predicted genes. The integration analysis was performed separately for upregulated and downregulated DE-miRNAs.

### 2.4. Gene Ontology Enrichment and KEGG Analysis

The protein–protein interaction (PPI) network for DE-miRNAs’ predicted target genes was identified using the Search Tool for the Retrieval of Interacting Genes/Proteins (STRING) online database (http://stringdb.org/) (accessed on 4 March 2022) [26]. Gene Ontology (GO) functional annotation for biological process and Kyoto Encyclopedia of Genes and Genomes (KEGG) pathway enrichment analysis were performed for integrated target genes of DE-miRNAs. A *p*-value of <0.05 was regarded as statistically significant.

### 2.5. Identification and Analysis of Hub Gene

The protein–protein interaction (PPI) for DE-miRNAs from STRING database was exported to Cytoscape software (version 3.9) and visualized [27]. The top 30 hub genes were selected (the top 30 nodes of the PPI network) using the Maximal Clique Centrality (MCC) method [28], which has a better performance in terms of its precision in predicting the top essential proteins. Further analysis was performed using ClueGO [29] to integrate GO terms as well as KEGG pathways and create a functionally nested or organized GO/pathway term (k score = 3). This task analyzes one set of genes or compares two lists of genes and comprehensively visualizes functionally grouped terms [29].

### 2.6. Real-Time Polymerase Chain Reaction for Determination of mRNA Expression of Hub Genes

Total RNA extraction and complementary DNA synthesis were performed as previously described [30]. Briefly, testis samples for each animal were used to extract RNA by TRizol Invitrogen, Carlsbad, CA, USA) tissue homogenization method. The RNA concentration and quality were determined using a NanoDrop-1000 spectrophotometer (Thermo Scientific, Rockford, IL, USA), and all RNA samples were treated with DNAse I (Invitrogen) to remove the DNA contaminant. Complementary DNA was synthesized using the iScript cDNA synthesis kit (Bio-Rad Laboratories Inc., Hercules, CA, USA) from each biological replicate and stored at −20 °C.

Specific primer pairs (Table 2) for the hub genes were designed using primer-BLAST (www.ncbi.nlm.nih.gov/tools/primer-blast/, accessed on 10 January 2022). Prior to real-time PCR, ethidium bromide-stained electrophoresis gel for the amplicon of the expected size was performed (Appendix A). Real-time PCR for each sample was carried out using Fast SYBR Green Master Mix (Applied Biosystems, Foster City, CA, USA) as described [25] following the manufacturer’s instructions. Endogenous control glyceraldehyde-3-phosphate dehydrogenase (GADPH) was used to normalize the threshold cycle (CT) values. Fold comparisons were made between the mature and immature groups.

Statistical analysis was performed using SAS Analytics software (9.4 version; SAS Institute, Cary, NC, USA). A *p* value ≤ 0.05 is considered as statistically significant. Analysis of variance (ANOVA) was used to calculate statistical significance. All mRNA data are expressed as mean ± SEM. The correlation (r) between miRNA and mRNA was calculated using PROC CORR of SAS (Pearson correlation coefficient).

### 2.7. Protein Immunoblots

Western blots for each testis tissue sample from mature and immature dogs were performed separately by methods described previously [30]. Briefly, protein extraction methods included: the addition of protease and phosphatase inhibitor to the testis sample, homogenization, lysate incubation at 4 °C for 45 min, centrifugation (at 12,000× *g* for 20 min), and determination of protein concentrations. Protein lysates (60 μg/lane) were then electrophoresed through 12% SDS-PAGE gel (Bio-Rad Laboratories, Philadelphia, PA, USA) and then transferred onto a PVDF membrane (Bio-Rad Laboratories). Samples were incubated in 10% goat serum in PBS to block non-specific binding. After overnight incubation at 4 °C with primary antibodies [mouse monoclonal to DNMT1 (Catalog # MA5-16169) and rabbit polyclonal to PTEN (Catalog # 600-401-859) from Thermo Fisher Scientific, Waltham, MA, USA; and mouse polyclonal to actin (sc-47778) from Santa Cruz Biotechnology, Santa Cruz, CA, USA], membranes were washed in buffer containing 2% animal serum and 0.1% detergent. The membranes were then incubated in secondary antibodies [goat anti-mouse IgG-FITC for DNMTA1 and β-actin (sc-2010; Santa Cruz Biotechnology), and goat anti-rabbit IgG-FITC for PTEN (sc-2012; Santa Cruz Biotechnology) for 1 h at room temperature. The blots were then washed and scanned using the Pharos FX Plus system (Bio-Rad Laboratories). FITC fluorophore was excited at 488 nm and read at the emission wavelength of 530 nm. All possible negative controls, equivalent concentrations of nonspecific IgG or normal serum in place of the primary antibody, were included.

## 3. Results

For MiRNA-associated gene quantitative profiling, 32 upregulated and 12 downregulated miRNAs were included for in-silico analysis (Figure 1). Similarities of the nucleotide sequences of differentially expressed canine miRNAs were comparable with those of homo sapiens (Table 3). These up- and down-regulated miRNAs were submitted to elucidate predicted genes. Of 32 upregulated miRNAs submitted, 31 predicted 560 genes (Appendix A) and of 12 downregulated miRNAs, 11 predicted 53 genes (Appendix A).

Figure 2A shows the PPI for the upregulated miRNAs (546 nodes; 3134 edges; PPI enrichment *p* < 1.0 × 10^−16^) and reveals 978 significantly enriched biological processes GO terms (False Recovery Rate, *p* < 0.05) and 127 significant (False Recovery Rate, *p* < 0.05) KEGG enrichment pathways (Appendix A). Figure 2B shows the PPI for the down-regulated miRNAs (53 nodes and 138 edges, PPI enrichment *p* < 1.11 × 10^−14^) and reveals 405 significantly enriched biological processes GO terms (False Recovery Rate, *p* < 0.05) and 72 significant (False Recovery Rate, *p* < 0.05) KEGG enrichment pathways (Appendix A).

The top-ranked 30 hub genes using Maximal Clique Centrality (MCC) method for upregulated miRNAs and downregulated miRNAs were screened and presented in Figure 3A and 3B, respectively. Table 4 shows the hub genes and their roles, tissue expression, and protein–protein interactions (up to six closely related genes), for up-and down-regulated miRNAs in adult dog testis. Canine miRNAs and associated hub genes are presented in Table 5. Further, the top 30 upregulated and downregulated hub genes’ associated KEGG pathways are presented in Table 6.

To interpret functionally nested gene ontology and pathway annotation networks for the predicted genes of adult dog testis, ClueGo nested network analysis was performed, and the results are presented in Appendix A for up- (29 GO term groups and 23 KEGG pathway groups) and down-regulated miRNAs (8 GO term groups and 2 KEGG pathway groups), respectively. For up-regulated hub genes, a functionally grouped network with terms as nodes (Figure 4A), GO-pathway terms specific for genes (Figure 4B), and a chart with functional groups including specific terms (Figure 4C) is presented. Similarly, for down-regulated hub genes, functionally grouped networks with terms as nodes (Figure 5A), GO-pathway terms specific for genes (Figure 5B), and a chart with functional groups including specific terms (Figure 5C) is presented.

The mRNA expressions for CDKN1A, EGFR, JUN, NOTCH1, and PIK3R1 were greater (*p* < 0.05) in abundance in mature compared to immature dog testis (*p* < 0.05; Figure 6); whereas the mRNA expressions for DNMT1, PTEN, ESR1, and TIMP3 were lower in abundances in mature compared to the immature testis (*p* < 0.05; Figure 6). The mRNA expressions of hub genes for upregulated miRNA were in greater abundance and the mRNA expressions of hub genes for downregulated miRNA were in lower abundance (*p* < 0.05). There was a positive association for the miRNA–mRNA pair (Figure 7; r = 0.60; *p* < 0.05). The relative expressions of miRNA and associated hub gene mRNA in mature dog testis were also provided in Table 7.

Protein immunoblots were performed to recognize the PTEN, DNMT1, and β-actin (reference gene) proteins. The PTEN, DNMT1, and β-actin proteins were 47, ~180, and 42 kDa, respectively (Appendix A).

## 4. Discussion

In the present study using DE-miRNAs between immature and adult dog testis samples, we identified 613 genes involved in the regulation of testis development. The GO biological functional enrichment showed that genes were mainly enriched in regulation of cellular process, cellular response to stimulus, developmental process, cell population proliferation, cell death, cell differentiation, apoptotic process, and cell metabolic process. The KEGG pathway enrichment showed that genes were mainly enriched in cancer biology, PI3K-Akt signaling pathway, AGE-RAGE signaling pathway, p53 signaling pathway, cellular senescence, hormone signaling pathway and human papillomavirus infection. Furthermore, we performed bioinformatics analysis to identify the potential key genes based on random selection algorithm, GO semantic similarity, PPI network, and cluster analysis. The results showed the involvement of differentially expressed genes in growth, sexual development, the maintenance of gluconeogenesis and lipid metabolism, cell proliferation, Sertoli and spermatogonial stem cells division and growth, cell cycle (cell cycle progression at G1-S phase), maturation, cell survival, and apoptosis. These key genes functioned as the essential molecules that might have mediated the testis development process between the immature and adult dogs.

Micro RNAs critically regulate the proliferation and/or early differentiation of stem cell populations in testis [31]. In mouse, deletion of both miR-34b and miR-34c led to sterility, resulting in reduced sperm count, changed sperm morphology, and abnormal motility [32]. It has been cited that miR-34c-5p could be used as biomarkers of germ cell maturation [33,34,35]. The differences in expressions of cfa-miR-34b and cfa-miR-34c were vast between mature and immature testis in the current investigation. The cfa-miR-34b and cfa-miR-34c associated with hub genes *NOTCH1* and *MYC* regulates the cell cycle, cellular fate determination, cell proliferation, cell differentiation, and cellular apoptosis. MicroRNA-34b expression was associated with meiotic-specific cells from the murine testis [36] and miR-34c was highly expressed in pachytene spermatocytes and round spermatids when compared with testicular somatic cells and other tissues in adult mice [37]. In humans, miR-34b was downregulated in asthenozoospermic and oligoasthenozoospermic sperm compared with normal sperm, suggesting its function is critical beyond sperm production [38]. The enhanced expression of miR-34c in the germ cells during the later steps of spermatogenesis indicates its functional significance in meiosis and spermiogenesis. Retinoic acid receptor gamma is one of the target genes of miR-34c, indicating the critical role of retinoic acid signaling in the control of meiosis [39]. MicroRNA-34 is intrinsically linked to the p53 tumor suppressor gene and the established Wnt cascade [40]. Wnt/b-catenin signaling is essential for the regulation of spermatogenesis [41], and the gene p53 is important for the biogenesis of acrosome and nuclear shaping during spermiogenesis. Previous studies demonstrated that Sertoli proliferation and differentiation can be mediated through the Wnt/β-catenin signaling pathway [42,43], mTOR signaling pathway [44,45,46], and TGF-β signaling pathway [47,48]. PTEN, PI3K/AKT, and STAT signaling pathways were found to be involved in bull sperm cell apoptosis [49]. Recent study on analysis of miRNAs and target mRNAs between immature and mature bull testis showed enrichment during Sertoli proliferation and differentiation, and sperm apoptosis [50]. Further, several differentially expressed genes enriched in metabolic pathways were involved in fat metabolism, including fatty acid degradation, adipocytokine signaling, and PPAR signaling pathway [50].

In the current study, cfa-miR-7a was upregulated in adult testis and *EGF* was identified as one of its target genes. Further, this upregulated miRNA in adult testis-associated hub genes *CDH1* and *MET* interacted with *EGF* and *EGFR*. The ErbB signaling pathway-associated genes found were *CDKN1A*, *EGFR*, *JUN*, *KRAS*, *MYC*, and *PIK3R1*. The *CDKN1A* is the primary p53 target gene that mediates cell-cycle arrest [51,52]. LH signaling was reduced in *CDKNIA* knockout mice plausibly affecting pubertal development [53]. The *EGF* mediates spermatogonial proliferation through its receptors *ErbB1*, *ErbB2*, and *ErbB4* in the testis [54]. It is possible that *EGF* mediates spermatogonial proliferation through its receptors on Sertoli cells via activation of MAPK cascade and/or PI3K cascade by elevating the expressions of *SCF*, *Ig-NRG1*, and *EGFR*s (*ERBBR*s) [55,56]. Further, KRAS expression patterns showed preferential tissue activation suggesting different cellular functions [57,58]. Interestingly, cfa-miR-125a was downregulated in adult testis and its associated hub genes were *ERBB2* and *ERBB3*. Aberrant *EGFR* activation is a significant factor in the development and progression of multiple cancers [59] suggesting that a balanced expression is warranted in testis development.

In the current study, the upregulated cfa-miR-29c-associated gene was *PIK3R1*, and its signals are important for cell activities, including cell growth and division, migration, production of new proteins, transport of materials within cells, and cell survival. Studies suggest that PI3K signaling may be involved in the regulation of several hormones, including insulin. NF-kappaB and PI3K-Akt pathways are among PI3 K-associated pathways. PI3K-Akt pathway is an intracellular signal transduction pathway that promotes metabolism, proliferation, cell survival, growth, and angiogenesis in response to extracellular signals [60]. This is mediated through serine and/or threonine phosphorylation of a range of downstream substrates. The PI3K-Akt pathway engages in many stages of male reproduction, including the regulation of the hypothalamus–pituitary–gonad axis during spermatogenesis, the proliferation and differentiation of spermatogonia and somatic cells, and the regulation of sperm autophagy and testicular endocrine function in the presence of endocrine disrupting chemicals [61]. Further, the PI3K-Akt pathway is required for the stimulatory actions of *FSH* [62]. It should be noted that the activation of *PI3K* by *EGF* occurs via the association of the *p85* subunit of *PI3K* (*PIK3R1*) with the activated *EGFR* [63]. PIK3R1/p85a is the most abundant isoform in normal tissues [64] but its expression is reduced in cancer suggesting that it regulates cell proliferation. In the current study, cluster analysis revealed PI3K-Akt pathway was regulated by associated genes *BCL2L11*, *CCND1*, *CCNE1*, *CDK4*, *CDK6*, *CDKN1A*, *EGFR*, *FGFR1*, *JAK2*, *KDR*, *KRAS*, *MCL1*, *MET*, *MYC*, *PIK3R1*, *PTEN*, and *RELA*.

Current investigation revealed that upregulated canine miR-15a and miR-378 were associated with hub gene *VEGFA*, which regulates neovascularization and cord formation, and potentially acts through the PI3K pathway during testis morphogenesis [65]. Further, *VEGF* signaling regulates germ cell proliferation and promotes testicular regeneration via direct action on germ cells and the enhancement of vascularization. Associated genes *KDR*, *KRAS*, and *PIK3R1* in the VEGF signaling pathway in the current study, may regulate the testis morphogenesis and regeneration.

Regulation of testis development by hormones has been described in [66]. Cluster analysis in the current investigation revealed the involvement of associated genes *EGFR*, *JUN*, *KRAS*, and *MMP2* in the regulation of the GnRH signaling pathway. The *FSH* regulates Sertoli cell proliferation during fetal and early postnatal life by activating cAMP/PKA/ERK1/2 and PI3K/Akt/mTORC1 dependent pathways, and by increasing the transcriptional activity of *c-MYC* and *HIF2* and the expression of *CCND1* [67]. The *CCND1* is a hub gene for upregulated canine miRNAs 15a, 16, 19a and 20a, and *MYC* is a hub gene for upregulated miR-34c in mature testis in the current investigation, suggesting the involvement in the Sertoli cell proliferation as indicated in the previous studies. Insulin and *IGF1* regulate testicular functions by activating PI3K/Akt and ERK1/2 signaling pathways. Mice lacking INSR and I*GF1R* in Sertoli cells showed a 72% reduction in testis size and a 79% reduction in daily sperm production [68,69]. Relaxin is another member of the insulin-related peptide family involved in Sertoli cell proliferation [66,70]. Associated genes *EGFR*, *JUN*, *KRAS*, *MMP2*, *PIK3R1*, and *RELA* engage in the relaxin signaling pathway in the current study, could have contributed to Sertoli cell proliferation. Genes *IGF1*, *AMH*, hedgehog (*DHH*), and platelet-derived growth factor (*PDGF*) seem to regulate Leydig cell differentiation and function. The *IGF1* stimulated differentiation and mitosis of Leydig cells [71]. Equally, the decrease in estrogen production inhibited Leydig cell differentiation in prepubertal and adult rat testes [72]. Since *IGF1*, *DHH*, and *PDGF* are Sertoli cells paracrine factors, it seems reasonable to speculate that thyroid hormone actions on Leydig cells might be, at least in part, mediated through Sertoli cells. Relaxin-induced Sertoli cell proliferation involves the activation of a Gi protein and the activation of EKR1/2 and PI3K/Akt pathways [66,70]. Activin A along with *FSH* regulates Sertoli cell proliferation during the fetal and postnatal period via the SMAD pathway [73,74]. The *NOTCH1* is identified as a hub gene for upregulated cfa-miR-34c in the current investigation. *NOTCH1*-IC forms a transcriptional complex with *SMAD*, a component of established TGF-β signaling, and regulates the expression of *HES1* by binding to the promoter [75]. These signal cross-talks are sophisticated regulation processes during cell fate determination and cancer development [76]. Interestingly, Nemo-like kinase (*NLK*) was a hub gene for downregulated miR-181a, -181b. The *NLK* phosphorylates the NOTCH1 protein. *NLK*-mediated phosphorylation does not interfere with the nuclear localization of *NOTCH1*-IC but decreases the association of the Notch active transcription complex [77]. Inhibin B is the main circulating inhibin produced by Sertoli cells; however, inhibin B has no role by itself but plays a role in the modulation of activin A-induced Sertoli cell proliferation [66,67].

Cytokines play an important regulatory role in the development and normal function of the testis. They signal via the adapter protein MyD88 to activate NFκB; and *TGFβ*s and activins, which signal through serine/threonine kinase receptor subunits to activate *SMAD* transcription factors. It has been shown that *IL1α* and *IL1β* increased DNA synthesis and Sertoli cell number in vitro and *IL1α* had a more potent effect than *IL1β* [78]. *TNFR1* has been detected in Sertoli cells and probably mediates *TNFα* biological actions, pro-inflammatory and immunoregulatory responses, and apoptosis [79]. A chemokine, *CXCL8*, was identified as one of top 30 hub genes in the current study. It should be noted that *PTEN* (hub gene for upregulated miR-22 and downregulated miR-214) loss induces a selective upregulation of *CXCL8* signaling that sustains cell growth and survival [80]. In the current study, cluster analysis revealed involvement of T and B cell receptor signaling pathway regulated by *CDK4*, *JUN*, *KRAS*, *PIK3R1*, and *RELA* genes. This regulation could have contributed to the development and normal function of the testis.

Androgen-dependent regulation of Sertoli cell proliferation is an indirect effect probably exerted through the secretion of a paracrine factor [66,67]. Direct effects of androgens on Sertoli cells seem to be related to maturation of this cell type. Estrogens play important roles in the regulation of testis development and spermatogenesis. Associated genes *EGFR*, *ESR1*, *JUN*, *KRAS*, *MMP2*, and *PIK3R1* were involved in the estrogen signaling pathway in the current study. ESR was a hub gene for miR-18a. -19a, and -22 in the current study. Estrogens increase proliferation of Sertoli cells through *ERα* and *GPER* and on the other hand, at the end of the proliferative period it promotes cessation of proliferation and cell maturation through *ERβ* [66,67]. *ERα* promotes cell proliferation through the activation of *NFκB* in a *PI3K*- and a *ERK1/2*-dependent manner and that this is accompanied by *CCND1* induction [81]. *GPER* activates Src/PI3K/Akt pathway which participates in E2-induced Sertoli cell proliferation via regulating the expression of S-phase kinase-associated protein 2 (SKP2) [82]. The *ERβ* promotes cell cycle exit and cell maturation through the activation of *CREB* in a PI3K-dependent manner and this leads to the expression of the Sertoli cell differentiation markers—*CDKN1B*, *GATA1* (isoform *GATA6* was a hub gene for down-regulated cfa-miR-181a and -181b), and *DMRT1* [66,67]. Role of thyroid hormone and retinoic acid in the cessation of proliferation and in maturation of Sertoli cells. Thyroid hormone regulated Sertoli cell maturation is through *TRα1* and *AIMP1* (*p43*) receptors. The mechanisms participating in these processes involve the regulation of *CX43*, *c-MYC* (*MYC* was identified as a hub gene for upregulated cfa-miR-34b and *MYCN* a hub gene for miR-101), *P21CIP1* (CDKN1A a hub gene for upregulated cfa-miR-106a), and *P27KIP1* (*CDKN1B* a hub gene for down-regulated cfa-miR-181a) [83,84,85]. Cluster analysis revealed associated genes *CCND1*, *ESR1*, *FOXO1*, *KRAS*, *MYC*, *NOTCH1*, and *PIK3R1* involved in the regulation of the thyroid signaling pathway.

Studies observed positive (upregulated–upregulated; complement) or negative (upregulated–downregulated; reverse complement) associations between miRNA and mRNA pairs [86]. It should be noted that miRNAs can activate gene expression directly or indirectly in response to different cell types and environments and in the presence of distinct cofactors. The biological outcome of miRNA–mRNA interaction can be altered by several factors contributing to the binding strength and repressive effect of a potential target site. Interestingly, it has been reported that significant miRNA–mRNA associations were complementing in normal tissue and miRNA–mRNA whereas associations were reverse complementing in the same tissue in diseased conditions [87]. The association between miRNA and hub gene mRNA expressions was positive in the current study.

Collectively, in the current study, GO analysis of differentially expressed genes showed that the down-regulated genes were significantly enriched in a large number of GO terms. Further, the GO analysis showed that most GO terms were downregulated, indicating that these DEGs may have played important roles in the development of the testis of immature dogs. However, the upregulated genes enriched in the GO terms were related to meiosis, protein ubiquitination, and fertilization of adult dogs. This suggests that a large number of genes with key roles in male reproduction traits are highly expressed in mature testis. Interestingly these genes are not expressed or expressed in low abundance in immature dog testis. Similar results were reported in a bull study [50]. Our analysis identified differentially expressed genes and differentially expressed miRNAs associated with male reproduction and elaborated cluster networks between miRNAs and genes regulating testis structure and function. This outcome provides significant insights into the molecular mechanisms of male fertility and spermatogenesis and will be valuable for future genetic and epigenetic studies of testis development and maturity.

## 5. Conclusions

The complex relationship between miRNA and gene interaction is a vital component of miRNA functional analysis in the testis development process. Abnormal expression of miRNAs and/or any regulation disturbance can lead to impaired germ cells, abnormal spermatogenesis, and even neoplasia. The present in-silico analysis showed the involvement of canine testicular miRNAs in structure and function. The DE-miRNAs between immature and adult canine testis and their associated genes involved in several regulatory pathways play a crucial role during the immature testis transition to the adult testis. A focused study of individual miRNA molecules may elucidate specific functions or problems, endorse the development of anti-oncogenic reagents and infertility/subfertility treatments, and bolster novel contraceptive technologies.

## Figures and Tables

**Figure 1 animals-13-01520-f001:**
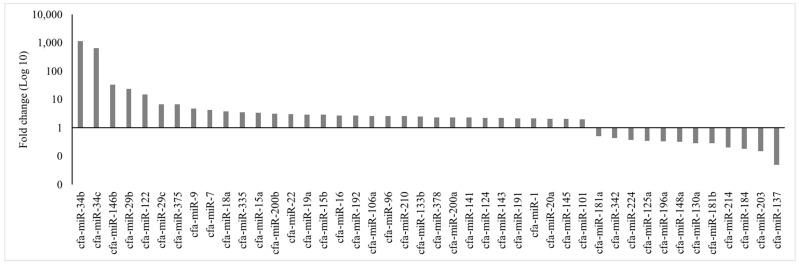
Up- and down-regulated microRNAs in adult canine testis compared with immature testis used for in silico analysis.

**Figure 2 animals-13-01520-f002:**
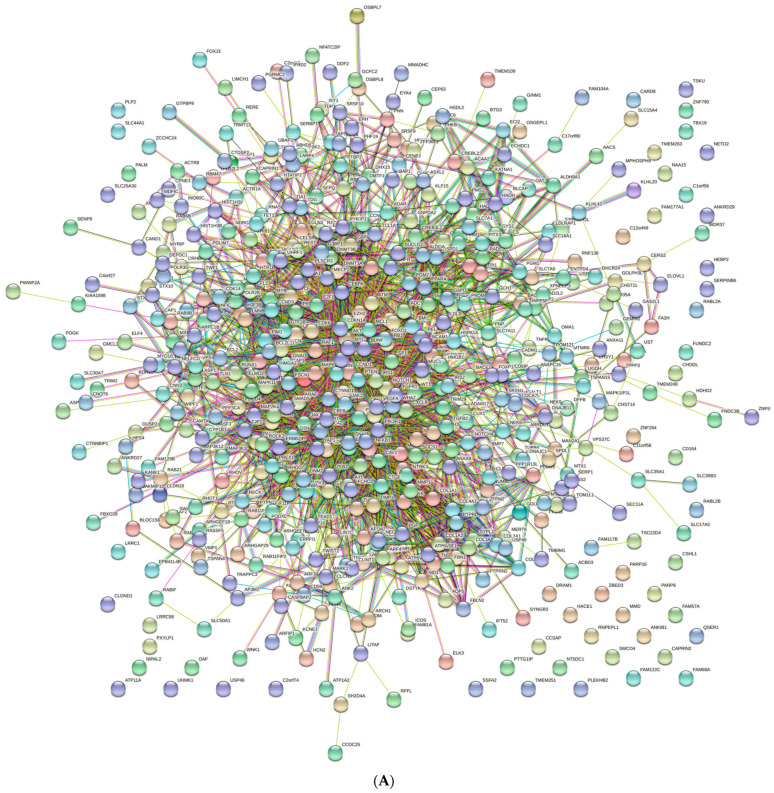
STRING protein–protein interaction (PPI) network. (**A**) PPI network of predicted genes (74) for the upregulated miRNAs (72 nodes and 281 edges, PPI enrichment *p* < 1.0 × 10^−16^). (**B**) PPI network of predicted genes (123) for the downregulated miRNAs (120 nodes and 189 edges, PPI enrichment *p* < 1.11 × 10^−14^). The color nodes represent proteins. The edges represent interactions.

**Figure 3 animals-13-01520-f003:**
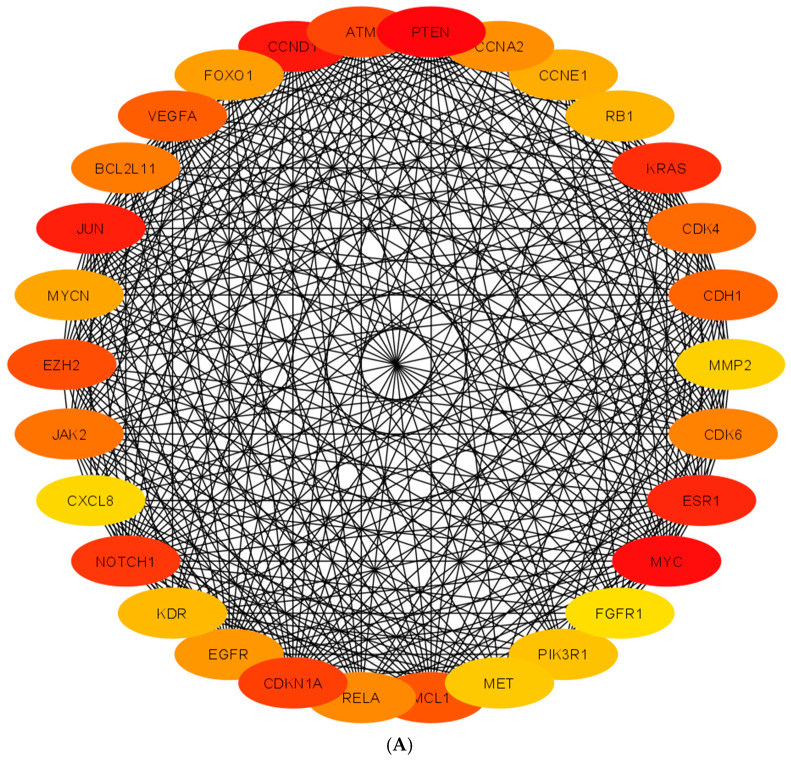
Protein–Protein Interaction (PPI) network of hub genes of DE-miRNAs. (**A**) PPI network of top genes for highly upregulated miRNAs. (**B**) PPI network of the top genes for downregulated miRNAs. DE-miRNAs are differentially expressed microRNAs; Black lines indicate interactions between genes. The PPI among hub genes for upregulated miRNAs was greater compared with hub genes for down-regulated miRNAs. The color red to yellow denotes a high to a low degree of expression. Black lines indicate interactions between genes.

**Figure 4 animals-13-01520-f004:**
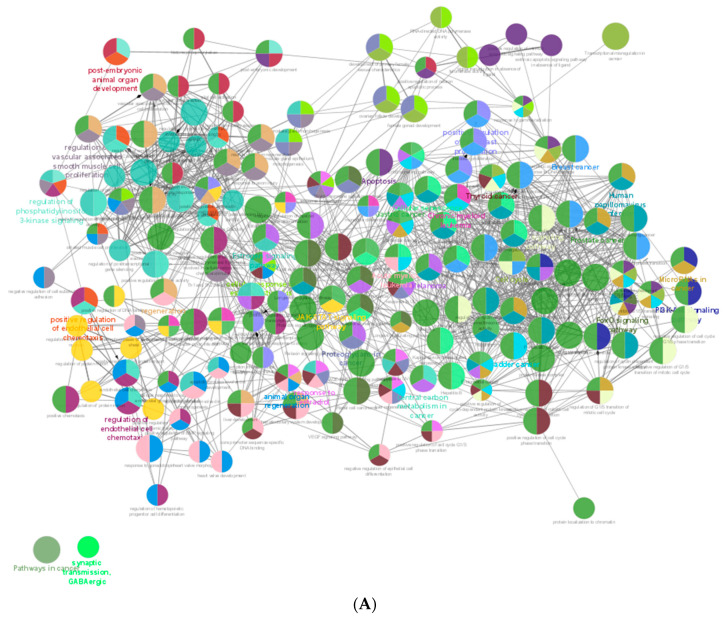
ClueGO analysis of upregulated hub genes in adult dog testis. (**A**) Functionally grouped network with terms as nodes linked, based on their kappa score levels (≥0.4), where only the label of the most significant term per group is shown. The node size represents the term enrichment significance. Functionally related groups partially overlap. The color gradient shows the gene proportion of each cluster associated with the term. (**B**) GO-pathway terms specific for upregulated genes. The bars represent the number of genes associated with the terms. The percentage of genes per term is shown as a bar label. (**C**) Overview chart with functional groups including specific terms for upregulated genes. The color gradient shows the gene proportion of each cluster associated with the term. (** *p* < 0.001).

**Figure 5 animals-13-01520-f005:**
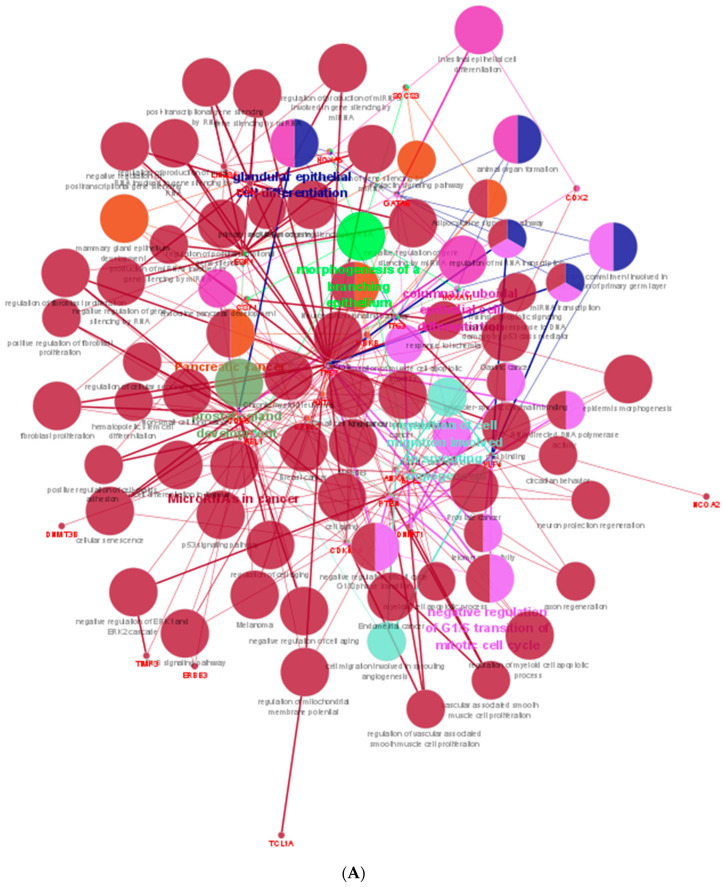
ClueGO analysis of downregulated hub genes in adult dog testis. (**A**) Functionally grouped network with terms as nodes linked, based on their kappa score levels (≥0.4), where only the label of the most significant term per group is shown. The node size represents the term enrichment significance. Functionally related groups partially overlap. The color gradient shows the gene proportion of each cluster associated with the term. (**B**) GO-pathway terms specific for upregulated genes. The bars represent the number of genes associated with the terms. The percentage of genes per term is shown as a bar label. (**C**) Overview chart with functional groups including specific terms for upregulated genes. The color gradient shows the gene proportion of each cluster associated with the term. (** *p* < 0.001).

**Figure 6 animals-13-01520-f006:**
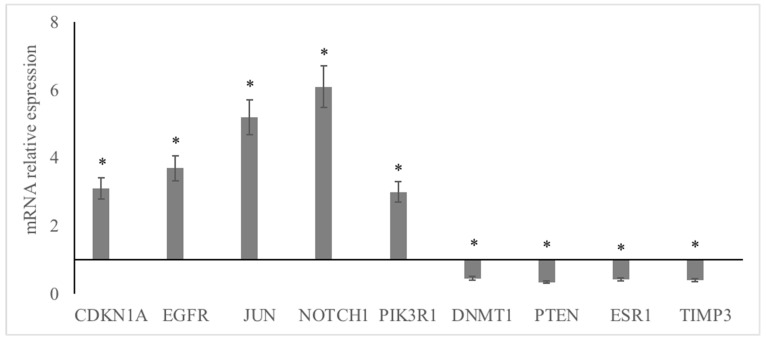
mRNA expression of hub genes in mature and immature canine testis. * Relative to immature dog testis (mRNA expression 1 fold) (*p* ≤ 0.05); *CDKN1A*, cyclin dependent kinase inhibitor 1A; *EGFR*, epidermal growth factor receptor; *JUN*, Jun proto-oncogene, AP-1 transcription factor subunit; *KRAS*, KRAS proto-oncogene, GTPase; *MYC*, MYC proto-oncogene, bHLH transcription factor; *PIK3R1*, phosphoinositide-3-kinase regulatory subunit 1; *GADPH*, glyceraldehyde-3-phosphate dehydrogenase (endogenous control).

**Figure 7 animals-13-01520-f007:**
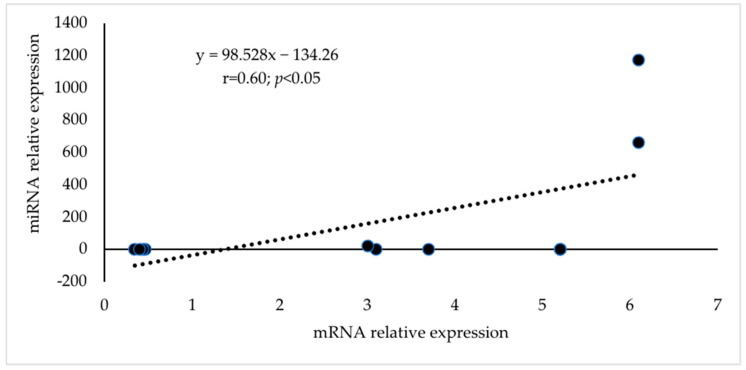
Correlation coefficient (r) of mean relative expressions of miRNA and mRNA pairs. ● Each point represents at least one miRNA-mRNA pair.

**Table 1 animals-13-01520-t001:** miScript canine miRNA polymerase chain reaction array.

Well	1	2	3	4	5	6	7	8	9	10	11	12
A	cfa-let-7a	cfa-let-7b	cfa-let-7c	cfa-let-7f	cfa-let-7g	cfa-miR-1	cfa-miR-101	cfa-miR-103	cfa-miR-106a	cfa-miR-106b	cfa-miR-10b	cfa-miR-122
B	cfa-miR-124	cfa-miR-125a	cfa-miR-125b	cfa-miR-126	cfa-miR-130a	cfa-miR-133a	cfa-miR-133b	cfa-miR-137	cfa-miR-141	cfa-miR-143	cfa-miR-145	cfa-miR-146a
C	cfa-miR-146b	cfa-miR-148a	cfa-miR-150	cfa-miR-15a	cfa-miR-15b	cfa-miR-16	cfa-miR-17	cfa-miR-181a	cfa-miR-181b	cfa-miR-182	cfa-miR-183	cfa-miR-184
D	cfa-miR-18a	Cfa-miR-191	Cfa-miR-192	Cfa-miR-195	Cfa-miR-196a	Cfa-miR-19a	Cfa-miR-200a	Cfa-miR-200b	Cfa-miR-200c	Cfa-miR-203	Cfa-miR-204	Cfa-miR-205
E	Cfa-miR-20a	cfa-miR-21	cfa-miR-210	cfa-miR-214	cfa-miR-218	cfa-miR-22	cfa-miR-222	cfa-miR-223	cfa-miR-224	cfa-miR-23a	cfa-miR-23b	cfa-miR-24
F	cfa-miR-25	cfa-miR-26a	cfa-miR-27a	cfa-miR-27b	cfa-miR-29b	cfa-miR-29c	cfa-miR-30b	cfa-miR-30c	cfa-miR-30d	cfa-miR-31	cfa-miR-335	cfa-miR-342
G	cfa-miR-34a	cfa-miR-34b	cfa-miR-34c	cfa-miR-375	cfa-miR-378	cfa-miR-451	cfa-miR-499	cfa-miR-7	cfa-miR-9	cfa-miR-92a	cfa-miR-93	cfa-miR-96
H	cel-miR-39-3p	cel-miR-39-3p	SNORD61	SNORD68	SNORD72	SNORD95	SNORD96A	RNU6-2	miRTC	miRTC	PPC	PPC

**Table 2 animals-13-01520-t002:** Forward and reverse primer sequence for quantitative real-time polymerase chain reaction amplification of mRNA for canine testis samples.

Gene	Primer Sequence (5′–3′)	Product Length	Accession Number
*CDKN1A*	F: CCTCGGAGGAGGTGCCAT	187	XM_038683340.1
	R: CGTCTCGGTGACGAAGTCAA		
*EGFR*	F: TAGGATCAGGGCCCGCAG	187	XM_038423676.1
	R: GCAACTTCCTGGATGGTCTTT		
*JUN*	F: CCTTCTACGACGATGCCCTC	101	XM_038666089.1
	R: GTTCAGGGTCATGCTCTGCT		
*NOTCH1*	F: CAGTGCAATGAGGGACCAGT	274	XM_038438708.1
	R: AGCATCCTCCACTCTCTGTCT		
*PIK3R1*	F: CACAACCTGCAAACATTGCC	160	XM_038659066.1
	R: AGGTCCCATCGGCTGTATC		
*DNMT1*	F: CTCTACGGTGTGTGCAGTGT	209	XM_038428673.1
	R: CAGGTGACCACGCTTACAGT		
*PTEN*	F: CATCATCAAGGAGATCGTCAGCAG	217	NM_001003192.1
	R: ATGTCTTTCAGCACACAGATTGTA		
*ESR1*	F: CACGGAGCTACACGCACAT	74	NM_001286958.2
	R: GGCTTGTAGAAGTCAAGGGCT		
*TIMP3*	F: CCTCCAAGAACGAGTGCCTT	161	NM_001284439.1
	R: GGGGTCTGTGGCATTGATGA		
*GAPDH*	F: AACATCATCCCTGCTTCCAC	234	NM_001003142.2
	R: GACCACCTGGTCCTCAGTGT		

*CDKN1A*, cyclin-dependent kinase inhibitor 1A; *EGFR*, epidermal growth factor receptor; *JUN*, jun proto-oncogene, AP-1 transcription factor subunit; *NOTCH1*, notch receptor 1; *DNMT1*, DNA methyltransferase 1; *PIK3R1*, phosphoinositide-3-kinase regulatory subunit 1; *PTEN*, phosphatase and tensin homolog; *ESR1*, estrogen receptor 1; *TIMP3*, tissue inhibitor of metalloproteinases 3; GADPH, glyceraldehyde-3-phosphate dehydrogenase.

**Table 3 animals-13-01520-t003:** Differentially expressed microRNAs sequence for dog and human.

miRNA ID	Sequence
cfa-miR-34b	AGGCAGUGUAAUUAGCUGAUUG
hsa-miR-34b	**U**AGGCAGUGUCAUUAGCUGAUUG
cfa-miR-34c	AGGCAGUGUAGUUAGCUGAUUGC
hsa-miR-34c	AGGCAGUGUAGUUAGCUGAUUGC
cfa-miR-146b	UGAGAACUGAAUUCCAUAGGCU
hsa-miR-146b	UGAGAACUGAAUUCCAUAGGCU**G**
cfa-miR-29b	UAGCACCAUUUGAAAUCAGUGUU
hsa-miR-29b	UAGCACCAUUUGAAAUCAGUGUU
cfa-miR-122	UGGAGUGUGACAAUGGUGUUUG
hsa-miR-122	UGGAGUGUGACAAUGGUGUUUG
cfa-miR-29c	UAGCACCAUUUGAAAUCGGUUA
hsa-miR-29c	**AUCUCUUACACAGGCUGACCGAUUUCUCCUGGUGUUCAGAGUCUGUUUUUGUC**UAGCACCAUUUGAAAUCGGUUA**UGAUGUAGGGGGA**
cfa-miR-375	GC**CCCGC**GACGAGCCCCUCGCACAAACC**GGACCUGAGCGUUUUGUUCGUUCGGCUCGCGUGAGGCAGGGG**
hsa-miR-375	GCGACGAGCCCCUCGCACAAACC
cfa-miR-9	UCUUUGGUUAUCUAGCUGUAUGA
hsa-miR-9	UCUUUGGUUAUCUAGCUGUAUGA
cfa-miR-7	UGGAAGACUAGUGAUUUUGUUGU
hsa-miR-7	UGGAAGACUAGUGAUUUUGUUGU**U**
cfa-miR-18a	UAAGGUGCAUCUAGUGCAGAUA
hsa-miR-18a	UAAGGUGCAUCUAGUGCAGAUA**G**
cfa-miR-335	UCAAGAGCAAUAACGAAAAAUGU
hsa-miR-335	UCAAGAGCAAUAACGAAAAAUGU
cfa-miR-15a	UAGCAGCACAUAAUGGUUUGU
hsa-miR-15a	UAGCAGCACAUAAUGGUUUGU**G**
cfa-miR-200b	CAUCUUACUGGGCAGCAUUGGA
hsa-miR-200b	CAUCUUACUGGGCAGCAUUGGA
cfa-miR-22	AAGCUGCCAGUUGAAGAACUGU
hsa-miR-22	AAGCUGCCAGUUGAAGAACUGU
cfa-miR-19a	UGUGCAAAUCUAUGCAAAACUGA
hsa-miR-19a	UGUGCAAAUCUAUGCAAAACUGA
cfa-miR-15b	UAGCAGCACAUCAUGGUUUA
hsa-miR-15b	UAGCAGCACAUCAUGGUUUA**CA**
cfa-miR-16	UAGCAGCACGUAAAUAUUGGCG
hsa-miR-16	UAGCAGCACGUAAAUAUUGGCG
cfa-miR-192	CUGACCUAUGAAUUGACAGCC
hsa-miR-192	CUGACCUAUGAAUUGACAGCC
cfa-miR-106a	AAAGUGCUUACAGUGCAGGUAG
hsa-miR-106a	**A**AAAGUGCUUACAGUGCAGGUAG
cfa-miR-96	UUUGGCACUAGCACAUUUUUGCU
hsa-miR-96	UUUGGCACUAGCACAUUUUUGCU
cfa-miR-210	**A**CUGUGCGUGUGACAGCGGCUGA
hsa-miR-210	CUGUGCGUGUGACAGCGGCUGA
cfa-miR-133b	UUUGGUCCCCUUCAACCAGCUA
hsa-miR-133b	UUUGGUCCCCUUCAACCAGCUA
cfa-miR-378	ACUGGACUUGGAGUCAGAAGGC
hsa-miR-378	ACUGGACUUGGAGUCAGAAGGC
cfa-miR-200a	CAUCUUACCGGACAGUGCUGGA
hsa-miR-200a	CAUCUUACCGGACAGUGCUGGA
cfa-miR-141	AACACUGUCUGGUAAAGAUGG
hsa-miR-141	**U**AACACUGUCUGGUAAAGAUGG
cfa-miR-124	UAAGGCACGCGGUGAAUGCCA
hsa-miR-124	UAAGGCACGCGGUGAAUGCCA**A**
cfa-miR-143	UGAGAUGAAGCACUGUAGCUC
hsa-miR-143	UGAGAUGAAGCACUGUAGCUC
cfa-miR-191	CAACGGAAUCCCAAAAGCAGCU
hsa-miR-191	CAACGGAAUCCCAAAAGCAGCU**G**
cfa-miR-1	UGGAAUGUAAAGAAGUAUGUA
hsa-miR-1	UGGAAUGUAAAGAAGUAUGUA**U**
cfa-miR-20a	UAAAGUGCUUAUAGUGCAGGUAG
hsa-miR-20a	UAAAGUGCUUAUAGUGCAGGUAG
cfa-miR-145	GUCCAGUUUUCCCAGGAAUCCCU
hsa-miR-145	GUCCAGUUUUCCCAGGAAUCCCU
cfa-miR-101	UACAGUACUGUGAUAACUGA
hsa-miR-101	UACAGUACUGUGAUAACUGA**A**
cfa-miR-137	UUAUUGCUUAAGAAUACGCGU
hsa-miR-137	UUAUUGCUUAAGAAUACGCGU**AG**
cfa-miR-203	GUGAAAUGUUUAGGACCACUAG
hsa-miR-203	GUGAAAUGUUUAGGACCACUAG
cfa-miR-184	UGGACGGAGAACUGAUAAGGGU
hsa-miR-184	UGGACGGAGAACUGAUAAGGGU
cfa-miR-214	ACAGCAGGCACAGACAGGCAGU
hsa-miR-214	ACAGCAGGCACAGACAGGCAGU
cfa-miR-130a	CAGUGCAAUGUUAAAAGGGCAU
hsa-miR-130a	CAGUGCAAUGUUAAAAGGGCAU
cfa-miR-181b	AACAUUCAUUGCUGUCGGUG
hsa-miR-181b	AACAUUCAUUGCUGUCGGUG**GGU**
cfa-miR-148a	UCAGUGCACUACAGAACUUUGU
hsa-miR-148a	UCAGUGCACUACAGAACUUUGU
cfa-miR-196a	UAGGUAGUUUCAUGUUGUUGGG
hsa-miR-196a	UAGGUAGUUUCAUGUUGUUGGG
cfa-miR-125a	UCCCUGAGACCCUUUAACCUGU
hsa-miR-125a	UCCCUGAGACCCUUUAACCUGU**GA**
cfa-miR-224	CAAGUCACUAGUGGUUCCGUUU
hsa-miR-224	**U**CAAGUCACUAGUGGUUCCGUUU**AG**
cfa-miR-342	UCUCACACAGAAAUCGCACCCGU
hsa-miR-342	UCUCACACAGAAAUCGCACCCGU
cfa-miR-181a	AACAUUCAACGCUGUCGGUGAG
hsa-miR-181a	AACAUUCAACGCUGUCGGUGAG**U**

Letter in bold denotes differences in sequences. Nucleotide sequences of differentially expressed canine and human miRNAs were retrieved from miRBase, (www.mirbase.org) (accessed 4 March 2022).

**Table 4 animals-13-01520-t004:** (A) Upregulated top 30 hub genes, and their roles, human tissue expression, and protein–protein interactions (up to 6 closely related genes); (B) Downregulated top 30 hub genes, and their roles, human tissue expression and protein–protein interactions (up to 6 closely related genes).

Top Hub Genes	Roles	Tissue Expressions	PPIs
**(A)**			
*PTEN*	Regulation of cell division and growth; Sertoli cell and spermatogonial stem cells	testis, prostate	CSNK2A1, NEDD4, PDGFRB, SLC9A3R1, SPOP, USP7
*CCNA2*	Regulation of cell cycle	testis, prostate	CDC6, CDK2, CDKN1A, CDKN1B, E2F1, SKP2,
*CCNE1*	Cell cycle regulation and progression; cell proliferation	testis, prostate	CDK2, CDC25A, CDKN1A, CDKN1B, FBXW7, SKP2,
*KRAS*	relays signals from outside the cell to the cell’s nucleus	testis, prostate	ARAF, CALM1, PIK3CG. RAF1, RALGDS. RASSF5
*VEGFA*	endothelial cell proliferation, promotion of cell migration, inhibition of apoptosis	testis, prostate	FLT1, KDR, DLL4, HIF1A, MYOD1, STAT3, RUNX2, MYC,
*CDK4*	Regulation of cell cycle progression, G1 phase	testis, prostate	CCND1, CCND2, CCND3, CDKN1A, CDNK1B, RB1
*CDH1*	Regulation of cell–cell adhesions, mobility and proliferation; spermatogenic stem cells and type A spermatogonia	testis, prostate	CBLL1, CDC27, CTNNA1, CTNNB1, CTNND1, EGFR
*MMP2*	Regulate space between cells, cell architecture.	testis, prostate	CCL7, COL1A1, COL5A1, TIMP2, TIMP3, TIMP4
*CDK6*	Regulation of cell cycle progression, G1 phase	testis, prostate	CCND1, CCND2, CDKN2A, CDKN2B, CDKN2C, RB1
*ESR1*	maintenance of gluconeogenesis and lipid metabolism; regulate cell proliferation; growth sexual development	testis, prostate	EP300, NCOA1, NCOA2, NR2F1, CREBBP, TRIM24,
*MYC*	Regulates cell cycle, and proliferation and apoptosis	testis, prostate	BRCA1, FBXW7, MAX, RAF1, RUVBL1, SMARCA4
*FGFR1*	Cell proliferation, differentiation, survival and migration	testis, prostate	FGF1, FRS2, GRB14, NCAM1, NEDD4, PLCG1,
*PIK3R1*	Cell proliferation and survival	testis, prostate	ERBB3, GAB1, GRB2, IRS1, KHDRBS1, PIK3CA
*MET*	Cell growth and survival	testis, prostate	CBL, EGFR, GAB1, GRB2, HGF, PLXNB1, SRC
*MCL1*	Regulates cell apoptosis	testis, prostate	BAD, BAX, BCL2L11, BIK, BMF, TPT1
*RELA*	Regulates all types of cellular processes, including cellular metabolism, chemotaxis	testis, prostate	CREBBP, HDAC1, NFKB1, NFKB2, NFKBIA, NR3C1
*CDKN1A*	Regulates cell cycle progression at G1-S phase	testis, prostate	CCND1, CDK2, CDK4, GADD45G, PCNA, TSG101
*EGFR*	Directs the behavior of epithelial cells; regulates cell migration	testis, prostate	EGF, GRB2, PTPN1, SHC1, SOS1, SRC
*KDR*	Promotes proliferation, survival, migration and differentiation of endothelial cells	testis, prostate	CDH5, SHC1, SHC2, SRC, VEGFA, VEGFC
*NOTCH1*	cellular fate determination, cell proliferation, cell differentiation and cellular apoptosis	testis, prostate	DTX1, FBXW7, JAG1, PSEN1, RBPJ, SMAD3
*CXCL8*	protein coding gene attracts neutrophils, basophils, and T-cells	testis, prostate	ACKR1, CCL5, CXCR2, RELA, SDC1, TNFAIP6
*JAK2*	protein coding gene regulates cell growth	testis, prostate	EPOR, PTPN1, IRS3, PTPN11, SH2B1, SOCS1, STAT5B
*EZH2*	Regulates cell fate determination	testis, prostate	DNMT1, EED, HDAC1, RBBP4, SUZ12, VAV1
*MYCN*	Regulates cell growth and division and apoptosis	testis, prostate	AURKA, EZH2, FBXW7, MAX, SP1, ZBTB17
*JUN*	Protein coding gene regulates cell proliferation and apoptosis	testis, prostate	ATF3, FOS, FOSL1, FOSL2, JDP2, MAPK8
*BCL2L11*	Regulates anti- and pro-apoptic regulators	testis, prostate	BCL2, BCL2A1, BCL2L1, BCL2L2, DYNLL1, MCL1
*FOXO1*	Protects cell from oxidative stress; regulates cell proliferation	testis, prostate	AKT1, AR, CREBBP, ESR1, SIRT1, YWHAZ,
*CCND1*	Regulates cell cycle progression at G1-S phase	testis, prostate	AR, CDK2, CDK4, CDK6, CDKN1A, CDKN1B
*ATM*	Regulates cell proliferation	testis, prostate	ABL1, AP1B1, BRCA1, FANCD2, TRFF1, TP53,
**(B)**			
*LIN28*	Posttranscriptional regulator of genes involved in developmental timing and self-renewal in stem cells	testis, prostate	DHX36, IGF2BP3, LARP1, L1TD1, ZCCHC11
*NLK*	Negative regulator in cell proliferation	testis, prostate	FAM222A, LEF1, MYB, PKM, MAP3K7, SMAD4
*GATA6*	Regulation of cellular differentiation and organogenesis	testis, prostate	CDK9, EP300, EGLN3, KLF2, NKX2-1, EP300,
*KRT5*	Regulation of cell structural framework	testis, prostate	ALOX12, EGFR, KRT14, LARP7, PKP2, SUMO2
*CDX2*	Regulation of cell growth and differentiation	testis, prostate	CREBBP, EP300, GSK3B, HNF1A, PAX6, RELA
*DNMT1*	Regulates DNA methylation; maintain a transcriptionally repressive state of genes in undifferentiated stem cells	testis, prostate	DNMT3B, HDAC1, PCNA, RB1, UHRF1, USP7
*TIMP3*	regulation of cell growth, cell death, angiogenesis, and invasion	testis, prostate	ADAM17, AGTR2, ASGR2, IFI30, MMP2, MMP3
*CDKN1B*	Oppose cell cycle progression; regulators of cell proliferation	testis, prostate	CCND1, CCND3, CCND2, CDK2, CDK4, STMN1
*NCOA2*	Regulates cell growth, development, and homeostasis	testis, prostate	AR, ESR1, NR3C1, PPARG, RXRA, VDR
*ESR1*	maintenance of gluconeogenesis and lipid metabolism; regulate cell proliferation; growth sexual development	testis, prostate	EP300, NCOA1, NCOA2, NR2F1, CREBBP, TRIM24,
*TP53*	regulates cell division by keeping cells from proliferating in an uncontrolled way	testis, prostate	BCL2L1, DAXX, HSPA9, HMGB1, MDM2, TOPORS
*HOXA5*	Regulates morphogenesis and differentiation	testis, prostate	ELAVL1, FOXO1, FOXA2, MEIS1, PRMT6, TWIST1
*BCL2*	Regulation of apoptosis	testis, prostate	BAD, BAX, BBC3, BCL2L11, BID, BIK
*AKT2*	Regulation of cell proliferation, growth and survival	testis, prostate	APPL1, CHUK, HSP90AA1, SH3RF1, TTC3, TCL1A
*TCL1A*	Co-activator of AKT kinases Enhances cell proliferation, stabilizes mitochondrial membrane potential and promotes cell survival	testis, prostate	AKT1, AKT2, EP300, FOS, JUN, JUNB
*KLF4*	Prevents differentiation of stem cells	ovary, uterus, placenta	CREBBP, CTBP1, EP300, HDAC2, KLF6, SP1
*SOCS3*	maintenance of cell shape and integrity	ovary, uterus, placenta	EGFR, IL6ST, JAK2, LEPR, PTPN11
*IKBKB*	Regulation of cell growth and apoptosis	testis, prostate	CDC37, CHUK, IKBKG, NFKB1, NFKB1A, TRAF2
*PTEN*	Regulation of cell division and growth; sertoli cell and spermatogonial stem cells	testis, prostate	CSNK2A1, NEDD4, PDGFRB, SLC9A3R1, SPOP, USP7
*TP63*	Regulation of epithelial morphogenesis, and adult stem/progenitor cell	testis, prostate	DAXX, HNRNPAB, HIPK2, ITCH, TP53, TP73
*ERBB2*	Regulation of cell membrane; regulates cell proliferation and anti-apoptosis	testis, prostate	EGFR, ERBB3, ERBB4, ERBIN, GRB2, PIK3R1
*ABL1*	Regualtes cell growth, survival, cell adhesion, cell migration; cytoskeleton remodeling	testis, prostate	CRK, ABI1, DOK1, NCK1, RAD51, RIN1
*PIK3CB*	cell adhesion; immune (PIK3) and inflammatory responses	testis, prostate	PIK3R1, PIK3R2, PIK3R3, HCK, IRS1
*CDCK6*	prevents cell proliferation and regulates negatively cell differentiation	testis, prostate	CDKN2A, CDKN2B, RB1, CDKN2D, CCND1, CCND3,
*E2F6*	regulation of DNA replication, DNA repair, mitosis, and cell fate.	testis, prostate	KDM5C, PCGF6, RING1, RYBP, TFDP1, TFDP2
*ELAVL1*	Anti-proliferation of cell, negatively affects meiotic division	testis, prostate	AGO2, CHEK2, HNRNPA1, IGF2BP1, RBM3, TNPO2
*HOXA11*	Regulates cell proliferation and differentiation	testis, prostate	FOXO1, HDAC1, HDAC2, MEIS1, PGBD3, YY1
*DNMT3B*	Regulation of DNA methylation	testis, prostate	DNMT1, DNMT3A, EED, EZH2, HDAC1, UBE2I

**Table 5 animals-13-01520-t005:** (A). Upregulated miRNAs and associated hub genes. (B). Downregulated miRNAs and associated hub genes.

miRNA	Hub Gene
**(A)**	
cfa-miR-1	*MET*
cfa-miR-7	*EGFR*
cfa-miR-9	*CDH1*, *FOXO1*
cfa-miR-15a	*CCND1*, *JUN*, *MCL1*, *VEGFA*
cfa-miR-15b	*CCNE1*
cfa-miR-16	*CCND1*, *CCNE1*, *FGFR1*, *JUN*, *MCL1*, *VEGFA*
cfa-miR-18a	*ESR1*
cfa-miR-19a	*CCND1*, *BCL2L11*, *ESR1*, *PTEN*
cfa-miR-20a	*CCND1*, *VEGFA*
cfa-miR-22	*ESR1*, *PTEN*
cfa-miR-29b	*MCL1*, *MMP2*, *PIK3R1*
cfa-miR-29c	*PIK3R1*
cfa-miR-34b	*MYC*, *NOTCH1*
cfa-miR-34c	*NOTCH1*
cfa-miR-96	*FOXO1*
cfa-miR-101	*ATM*, *EZH2*, *MCL1*, *MYCN*
cfa-miR-106a	*CDKN1A*, *RB1*, *VEGFA*
cfa-miR-124	*CDK4*, *CDK6*, *EZH2*, *RELA*
cfa-miR-143	*KRAS*
cfa-miR-145	*CCNA2*, *MYC*
cfa-miR-192	*RB1*
cfa-miR-375	*JAK2*
cfa-miR-378	*VEGFA*
**(B)**	
cfa-miR-125a-5p	*ERBB2*, *ERBB3*, *LIN28*, *TP53*
cfa-miR-130a	*CSF1*, *HOXA4*
cfa-miR-137	*CDK6*, *E2F6*, *KLF4*, *NCOA2*
cfa-miR-148a	*DNMT1*, *DNMT3B*
cfa-miR-181a	*BCL2*, *CDKN1B*, *CDX2*, *ELAVL1*, *ESR1*, *GATA6*, *HOXA11*, *NLK*
cfa-miR-181b	*CDX2*, *ESR1*, *GATA6*, *NLK*, *TCL1A*, *TIMP3*
cfa-miR-184	*AKT4*
cfa-miR-196a	*ANXA1*, *IKBKB*, *KRT5*
cfa-miR-203	*ABL1*, *SOCS3*, *TP63*
cfa-miR-214	PTEN
cfa-miR-342	DNMT1

**Table 6 animals-13-01520-t006:** KEGG pathways associated with the top 30 upregulated and down-regulated hub genes.

Upregulated Hub Genes Associated KEGG Pathway	Down Regulated Hub Genes Associated KEGG Pathway
ErbB signaling pathway	ErbB signaling pathway
Cell cycle	p53 signaling pathway
p53 signaling pathway	Neurotrophin signaling pathway
Mitophagy	Prolactin signaling pathway
PI3K-Akt signaling pathway	Adipocytokine signaling pathway
Apoptosis	
Longevity regulating pathway	
Cellular senescence	
VEGF signaling pathway	
Adherens junction	
JAK-STAT signaling pathway	
Th1 and Th2 cell differentiation	
T cell receptor signaling pathway	
B cell receptor signaling pathway	
GnRH signaling pathway	
Estrogen signaling pathway	
Prolactin signaling pathway	
Thyroid hormone signaling pathway	
Relaxin signaling pathway	
AGE-RAGE signaling pathway in diabetic complications	
Cancer	

**Table 7 animals-13-01520-t007:** Relative expressions of miRNA and associated hub gene mRNA in mature dog testis.

mRNA	miRNA	mRNA Relative Expression	miRNA Relative Expression
*CDKN1A*	cfa-miR-106a	3.1	2.63
*EGFR*	cfa-miR-7	3.7	4.35
*JUN*	cfa-miR-15a	5.2	3.43
*JUN*	cfa-miR-16	5.2	2.75
*NOTCH1*	cfa-miR-34b	6.1	1175.42
*NOTCH1*	cfa-miR-34c	6.1	662.79
*PIK3R1*	cfa-miR-29b	3	23.59
*DNMT1*	cfa-miR-148a	0.46	0.32
PTEN	cfa-miR-214	0.34	0.2
ESR1	cfa-miR-181b	0.43	0.29
*TIMP3*	cfa-miR-181b	0.4	0.29

mRNA and miRNA expression relative to immature dog testis (*p* ≤ 0.05).

## Data Availability

Data are available from the corresponding author on reasonable request.

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
