# Peer review of "In Silico Analysis of miRNA-Mediated Genes in the Regulation of Dog Testes Development from Immature to Adult Form"

_animals, 2023, doi:10.3390/ani13091520_

Round 1

Reviewer 1 Report

The manuscript of Kasimanickam, and  Kasimanickam (In silico analysis of miRNA-mediated genes in the regulation 2 of dog testes development from immature to adult form) deals with the evaluation of data published in 2015 (Kasimanickam, V.R.; Kasimanickam, R.K. Differential expression of microRNAs in sexually immature and mature canine tes-502 tes. Theriogenology 2015, 83, 394-398.).

Here they present a detailed analyses of the results produced before.

However, there are some inconsistencies in this study.

Point 1: Due to the fact that this study is based on a former one, details about the experimental settings to get the samples are completely missing and should be added to the manuscript. Point 2: lane 155: “All possible negative controls were included.” What does this mean? Please add the information. Point 3: Table 2: Sequences form human are added. Where are the data from? Point 4: Lane 158: “Yellow color denotes differences in sequences”: there is no yellow colour. Point 5: Fig. 2: indicate Figure 2A and 2B. Point 6: Fig. 4: indicate Figure 4A, 4B and 4C. Figure 4B: not readable. Point 7: Fig. 5: indicate Figure 5A, 5B and 5C. Point 8: lane 258: should this be “Figure 6”? Point 9: the discussion part just summarize the results. This part should be integrated into the results part where more detailed description of the results is missing and re-write the discussion part. A better comparison with already existing data and the meaning of the found results for the development of the testis in the dog is missing here completely. Point 10: the number of tables should be reduced to a minimum, since all date are presented in figures too. The tables should be given as supplemental material.

Author Response

Reviewer 1:

The manuscript of Kasimanickam, and  Kasimanickam (In silico analysis of miRNA-mediated genes in the regulation 2 of dog testes development from immature to adult form) deals with the evaluation of data published in 2015 (Kasimanickam, V.R.; Kasimanickam, R.K. Differential expression of microRNAs in sexually immature and mature canine testes. Theriogenology 2015, 83, 394-398.). Here they present a detailed analyses of the results produced before. However, there are some inconsistencies in this study.

Authors: Thanks for the review and constructive comments.

Point 1: Due to the fact that this study is based on a former one, details about the experimental settings to get the samples are completely missing and should be added to the manuscript. 

Authors: The descriptions were added as suggested. Refer to line 76-89. Included table 1 as well.

Point 2: lane 155: “All possible negative controls were included.” What does this mean? Please add the information. 

Authors: The information was added as suggested. Refer to line 207-209.

Point 3: Table 2: Sequences form human are added. Where are the data from?

Authors: It was already included in line 133. The information was also added below the table for clarification.

Point 4: Lane 158: “Yellow color denotes differences in sequences”: there is no yellow colour. 

Authors: Correct table with “bold letters” to denote sequence differences was added.

Point 5: Fig. 2: indicate Figure 2A and 2B. 

Authors: All figures were submitted as separate files. This might be possible problem with the journal while creating pdf file. However, the figure numbers were included.

Point 6: Fig. 4: indicate Figure 4A, 4B and 4C. Figure 4B: not readable. 

Authors: Caption is included in each corresponding figures. This might be possible problem with the journal while creating pdf file. I will check with editors.

Point 7: Fig. 5: indicate Figure 5A, 5B and 5C. 

Authors: Caption is included in each corresponding figures. All figures were submitted as separate files. This might be possible problem with the journal while creating pdf file. However, the figure numbers were included.

Point 8: lane 258: should this be “Figure 6”? 

This is figure 3. This might be possible problem with the journal while creating pdf file. We will check with editors to make sure the figures are in order.

Point 9: the discussion part just summarize the results. This part should be integrated into the results part where more detailed description of the results is missing and re-write the discussion part. A better comparison with already existing data and the meaning of the found results for the development of the testis in the dog is missing here completely. 

Authors: In the results we reported the findings and in discussion we included were pertinent information on the findings that we reported in the results. In the revised version, we have added pertinent information in the discussion section as suggested. Refer lines 330 to 334, 349 to 357, and 482 to 495.

Point 10: the number of tables should be reduced to a minimum, since all date are presented in figures too. The tables should be given as supplemental material.

Authors: Thanks for the suggestion. We believe that tables and figures are important and that is what makes this investigation interesting. Thus we plan to keep the tables and figures as originally structured.

Reviewer 2 Report

In a previous study by the authors, the repertoire of canine testis miRNAs was analyzed using PCR array technology. The results demonstrated significant differences in miRNA expression patterns between sexually immature and mature dog testes, providing fundamental information on miRNAs and their role in spermatogenesis and male infertility in dogs. This approach offers the first comparative profile of the miRNA transcriptome in prepubertal and adult canine testes using miRNA PCR array. The authors provided important miRNA-predicted gene datasets for young and adult dogs and used a new analysis method in this study. However, we note a lack of novelty.

We will reconsider the paper for publication in this journal once the author addresses the following concerns.

In Figure 1, the author presented the differentially expressed miRNAs from a previously published paper. The results showed 32 up-regulated and 12 down-regulated miRNAs. The author should provide details on how the differentially expressed miRNAs were identified. Additionally, the y-axis does not match the bar chart.

The highlight color in Table 2 was not able to appear.

Please clarify in Figure 2A whether it shows the up-regulated miRNA or the miRNA-regulated genes PPI.

Please provide high-quality versions of Figures 4A, 4B, 5A, as the current versions are in low resolution.

In Figure 7 and its associated table (Table # is missing), the miRNA-mRNA correlation coefficient from mature dog testis is shown. I am wondering if the results for young dog testis can be provided.

Author Response

Reviewer 2:

In a previous study by the authors, the repertoire of canine testis miRNAs was analyzed using PCR array technology. The results demonstrated significant differences in miRNA expression patterns between sexually immature and mature dog testes, providing fundamental information on miRNAs and their role in spermatogenesis and male infertility in dogs. This approach offers the first comparative profile of the miRNA transcriptome in prepubertal and adult canine testes using miRNA PCR array. The authors provided important miRNA-predicted gene datasets for young and adult dogs and used a new analysis method in this study. However, we note a lack of novelty. We will reconsider the paper for publication in this journal once the author addresses the following concerns.

Authors: Thanks for the review and constructive comments.

We have included the following statements, in lines 482 to 495, which reflect the importance of/need for this investigation.

In Figure 1, the author presented the differentially expressed miRNAs from a previously published paper. The results showed 32 up-regulated and 12 down-regulated miRNAs. The author should provide details on how the differentially expressed miRNAs were identified. Additionally, the y-axis does not match the bar chart.

Authors: We have included the table (Table 1) and description in materials and methods (Lines 76 to 89). In the “y” axis we provided the names of the representative miRNA.

The highlight color in Table 2 was not able to appear.

Authors: It may be the problem with creation of the file. Correct table with bold letters to denote sequence differences was added.

Please clarify in Figure 2A whether it shows the up-regulated miRNA or the miRNA-regulated genes PPI.

Authors: It is clarified, in lines 277 and 278, as suggested

Please provide high-quality versions of Figures 4A, 4B, 5A, as the current versions are in low resolution.

Authors: In the revised version, we included high resolution images as per journal requirements.

In Figure 7 and its associated table (Table # is missing), the miRNA-mRNA correlation coefficient from mature dog testis is shown. I am wondering if the results for young dog testis can be provided.

Authors: We like to clarify that young dog testis values were used as reference values to calculate relative expressions in mature dogs.

Round 2

Reviewer 2 Report

I appreciate the author's detailed responses to their questions and concerns. However, I remain concerned about the use of the Y-axis in Figure 1. It appears to show only representative miRNAs and is not aligned with the bar chart, potentially leading to a loss of accuracy. If the author chooses to use this method of displaying data, they should provide a clear explanation for the selection of representative miRNAs in the text and figure legend.

I suspect that the missing Y-axis values may be attributed to scale or axis limits during chart generation. Therefore, I suggest regenerating the chart as a simple solution.

Author Response

Explanation was provided in the attached file
